# Performing tympanometry using smartphones

Justin Chan [1✉], Ali Najafi[2], Mallory Baker[3], Julie Kinsman[3], Lisa R. Mancl[4], Susan Norton[3,4,5], Randall Bly [3,5✉] & Shyamnath Gollakota [1,2✉]

## Abstract

**Background** Tympanometry is used as part of a battery of tests for screening of middle ear function and may help diagnose middle ear disorders, but remains available only on expensive test equipment.

**Methods** We report a low-cost smartphone-based tympanometer system that consists of a lightweight and portable attachment to vary air pressure in the ear and measure middle ear function. The smartphone displays a tympanogram and reports peak acoustic admittance in realtime. Our programmable and open-source system operates at 226 Hz and was tested on 50 pediatric patient ears in an audiology clinic in parallel with a commercial tympanometer.

**Results** Our study shows an average agreement of 86 ± 2% between the 100 tympanograms produced by the smartphone and commercial device when five pediatric audiologists classified them into five classes based on the Liden and Jerger classification.

**Conclusion** Given the accessibility and prevalence of budget smartphones in developing countries, our open-source tool may help provide timely and affordable screening of middle ear disorders.

## Plain language summary

Tympanometry is a test used to evaluate the health of the middle ear, which is involved in hearing, and helps in diagnosing middle ear disorders. However, the test currently requires expensive equipment and is limited in availability in resource-constrained settings. We present a low-cost smartphone-based tympanometry system. The device is able to change the air pressure of the ear canal and measure ear drum mobility. Our system, for which the software and hardware are openly available, is tested in 50 ears from children attending an audiology clinic. A panel of pediatric audiologists classified tympanometry measurements from our device and a commercial tympanometer with good agreement. Given the increasing availability of smartphones in developing countries, our system has the potential to make screening of middle ear disorders more accessible.

[1] Paul G. Allen School of Computer Science and Engineering, University of Washington, Seattle, WA, USA. [2] Department of Electrical and Computer Engineering, University of Washington, Seattle, WA, USA. [3] Seattle Children's Hospital and Research Institute, Seattle, WA, USA. [4] Department of Speech & Hearing Sciences, University of Washington, Seattle, WA, USA. [5] Department of Otolaryngology - Head and Neck Surgery, University of Washington, Seattle, WA, USA. ✉email: jucha@cs.washington.edu, Randall Bly randall.bly@seattlechildrens.org; gshyam@cs.washington.edu

Tympanometry is an objective test of middle ear function and can be used in combination with other tests like otoscopy and pneumatic otoscopy to diagnose middle ear disorders in accordance with clinical guidelines[1,2]. Accurate screening for middle ear disorders can result in more timely referrals to specialists and could contribute to the reduction of serious complications[3,4]. During a tympanometry test, the air pressure is changed in the ear canal to evaluate the mobility of the tympanic membrane and ossicular chain[5]. Although this core principle behind tympanometry was introduced in the 1950–1960s[6], there has not been many major advances in making these devices affordable and accessible for resource-constrained settings. As a result, existing tympanometers remain expensive, ranging from $2000 (handheld Otowave, Amplivox) to $5000 (AutoTymp, Welch Allyn), limiting its availability in resource-constrained scenarios including rural areas and developing countries[7].

At the same time, smartphone technology has advanced substantially over the last two decades. Today, budget smartphones that cost $40–50 second-hand contain powerful processors, user interfaces and high-resolution displays[8]. Here, we present an inexpensive tympanometry system that uses a lightweight and portable attachment to smartphones. It is designed to automatically detect when a seal has been formed with an ear canal, safely vary air pressure, and generate a tympanogram on the smartphone in real-time. Our attachment can be assembled using electronic and passive components with a material cost of $28. Due to the prevalence of smartphones in developing countries[9–11], our tympanometry tool may help increase timely access to otologic healthcare. We open-source our hardware and software making our system free and accessible to use and further adapt.

## Methods

**Concept and Prototype.** We designed our smartphone tympanometer in two different form factors. Figure 1a and Supplementary Fig. 1 shows a handheld design that holds all the electronics in a 3D printed plastic enclosure that attaches to the back of a smartphone. A handheld form factor may be suitable for use in mobile health clinics[12–14] or as part of medical humanitarian trips (e.g., doctors without borders)[15,16]. The desktop form factor shown in Fig. 1b mimics the design of existing commercial tympanometers. All electronic components of the system except for the smartphone are assembled onto a 42 × 45 mm custom printed circuit board (PCB) (Fig. 1c, Supplementary Figs. 7–11).

The pressure and acoustic sensors are connected to the ear probe through 1 m of lightweight air-tight silicone tubes to provide greater mobility during measurement. These silicone tubes are protected from damage using a flexible cable sleeve. The ear probe is designed to be lightweight and can rest securely in a patient's ear during a measurement without any additional applied force. Additionally, the ear probe was designed to be compatible with existing rubber ear tips (Grason & Associates) used with the commercial tympanometers in our clinical study[17]. Our tympanometry platform is easily programmable on the smartphone where parameters such as pressure speed, pressure limits, tone frequency, and volume can be modified in the software app (see Supplementary Fig. 2).

The smartphone attachment has multiple low-cost components. It consists of a pressure transducer made from a stepper motor to precisely move the plunger of a 5 mL syringe. Feedback from an onboard pressure sensor is used to accurately change the pressure between −400 and 200 daPa[5]. Figure 1d shows the syringe plunger moving a total distance of 5.3 mm to cover the pressure range. During this pressure sweep, the system sends a 226 Hz audio tone at 85 dB SPL and records the acoustic reflections at a microphone connected to the smartphone by a 3.5 mm audio jack. A frequency of 226 Hz was used in our clinical study, as it is the recommended frequency for patients over 9 months of age[18], though our system can be programmed to emit other frequencies. When the measurement is complete, the digital pressure data is sent to the smartphone using an onboard wireless Bluetooth radio. Figure 3a, b show the acoustic data bandpass filtered to 220–230 Hz and the pressure data from a single measurement. To ensure safety, our system is designed to terminate the measurement if the syringe of diameter 12.5 mm has moved forward by more than 16 mm. This termination condition was added to avoid large changes in air pressure in the ear. It is a fail-safe in the event of malfunction with the pressure sensor. We note that during our study there were no failures with the pressure sensor.

Our real-time algorithms run on the microcontroller to detect when a seal is formed within the ear canal and automatically begin measurement, without any user intervention. To detect if there is a seal, the microcontroller moves the syringe plunger back and forth. When the probe tip is outside the ear, only small changes in pressure at an average rate of 2.03 daPa/s are detected (Fig. 3c). If the ear probe has an adequate seal, the change in air pressure will exceed a predefined threshold (17 daPa/s). Our algorithm begins the measurement automatically by sweeping the pressure from 200 to − 400 daPa and then returning to 0 daPa.

If the probe is dislodged from the ear canal during a measurement, air pressure returns to ambient pressure. Our algorithms use this to detect that the seal has been lost (Supplementary Fig. 3, Supplementary Table 2). Our smartphone algorithm also detects if the eartip is occluded which can occur if it is flush against the wall of the ear canal or obstructed with cerumen. Since pressure changes in a clear and occluded ear canal are similar, our algorithm instead checks for high amplitude acoustic reflections as an indicator for occlusion (Supplementary Fig. 4).

In a healthy patient, an increase in positive and negative pressure tense the eardrum and make it more stiff, which causes an increase in reflected sound energy. The tympanogram is generated on the smartphone by converting the sound energy into calibrated units of acoustic admittance magnitude (Fig. 3d). The calibration process is performed by comparing reflections from the ear to reflections generated from cavities of known volumes.

Ears with middle ear fluid may have a stiff eardrum and the amount of reflected sound energy at different pressure levels may not have a pronounced peak. Patients with ear tubes or a tympanic membrane perforation will likely have a consistently attenuated reflection at different pressures that would indicate an unusually large ear canal volume. Middle ear conditions that affect the eardrum, like chronic eustachian tube dysfunction with a retracted eardrum and without fluid, may result in the tympanogram peak occurring at a negative pressure. An overly stiff eardrum, in the case of reduced ossicular mobility such as otosclerosis, may cause a lower than normal peak amplitude and an ossicular discontinuity may cause a higher than normal peak amplitude[5]. The tympanogram output displayed on the smartphone (see Supplementary Fig. 2) can be interpreted in the same way as it is on commodity tympanometers.

**Hardware design.** The electronic components of our hardware (Fig. 1a, b and Supplementary Fig. 1) can be divided into three groups: acoustic, pressure, and computing components. Our acoustic hardware consists of a printed circuit board (PCB) (Fig. 1c) with a speaker (Knowles SR-32453-000, $4.42) and

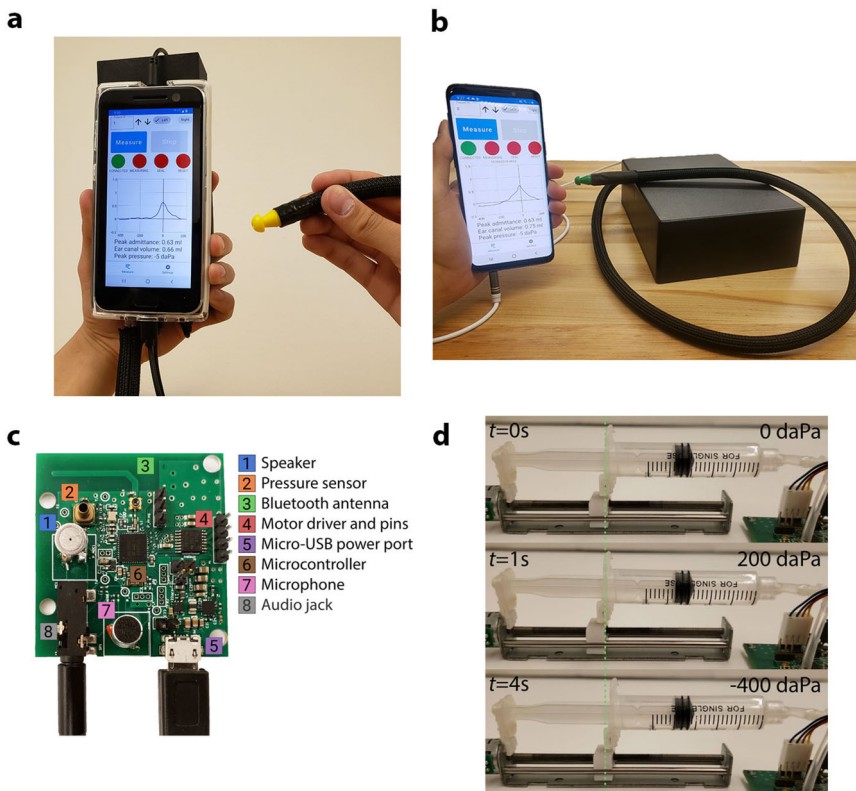

**Fig. 1 Low-cost smartphone-based tympanometer. a** All components of the hand-held tympanometer fit into a portable 3D printed enclosure that attaches to the back of a smartphone. **b** A desktop version of our tympanometer. **c** Close-up of PCB containing the key acoustic and pressure sensing elements of the tympanometer. The PCB includes a microcontroller, Bluetooth antenna and micro-USB port for computing, communication, and power respectively. **d** Through precise movements of a plunger and syringe, the stepper motor introduces positive and negative pressure into the ear canal. The green line is a reference marker for the plunger location at the beginning of the measurement. The plunger moves 5.3 mm to perform the pressure sweep from 200 to −400 daPa.

microphone (DB Unlimited MO064404-1, $0.29) to send and receive a 226 Hz audio tone. The speaker and microphone are connected directly to a smartphone via a 3.5 mm headphone jack. The smartphone sends a tone to the speaker with a sound level of 85 dB SPL at the ear canal, and records audio from the microphone at a sampling rate of 24 kHz. Our pressure hardware consists of a linear stepper motor (DC 2-Phase 4-wire Stepper Motor, $6.16) connected to a syringe and plunger (Frienda syringe, $0.55). The stepper motor is driven by a motor driver (Toshiba TB6612FNG, $0.90). A pressure sensor (Honeywell MPRLS0025PA00001A, $3.97) reads air pressure values in the ear canal. Finally, a microcontroller (Nordic Semiconductor nRF52832, $2.28) continuously monitors the air pressure values and runs the algorithm to check if there is a seal between the ear tip and ear canal, and performs the pressure sweep from 200 to −400 daPa (Fig. 1d). The microcontroller sends the recorded pressure values to the smartphone over Bluetooth at the end of the measurement to compute the tympanogram (Fig. 2). When the measurement is complete, the plunger of the syringe is automatically moved back to its original position. The PCB and its components can be powered by a smartphone through a micro-USB to USB-C cable. The current draw of the PCB when the motor is in use is 400 mA, which is within the current draw specifications for USB power delivery 2.0 used by USB type-C connectors[19].

Next we describe the set of passive components used to couple the acoustic and pressure components to the ear. There are three main connections. First, the microphone, speaker, syringe and pressure sensor are each directly connected to thin (1 mm ID,

2 mm OD) and flexible silicone tubing. Second, to group the connections, the three silicone tubes are encased in a larger (3 mm ID, 4 mm OD) silicone tube. Finally, the grouped silicone tube is attached to the probe head, which is a plastic adapter that can interface with standard tympanometer ear tips. All the components in the desktop version of our system are enclosed in a plastic enclosure (Hammond Manufacturing 1591XXFSFL, 221.01 × 150.01 × 63.50 mm). The total material cost of all electronic and passive components in the system is around $25 (Supplementary Table 1).

**Smartphone application**. A custom smartphone application was created to work with the Android operating system (Supplementary Fig. 2). The application first waits for user input to begin the measurement session. Upon confirmation, the application sends a Bluetooth packet to the microcontroller to begin checking if the ear tip has formed an airtight seal with the ear canal. Whenever a seal has been found or lost (Supplementary Fig. 3), the microcontroller sends a Bluetooth notification to the smartphone, which displays this in the form of a notification to the user. The application is responsible for sending the acoustic tone to the microphone, and for recording the reflected sound energy at the microphone. When the measurement session is complete the application will receive a confirmatory Bluetooth packet from the microcontroller along with a list of timestamped pressure values.

**Algorithm to identify a seal**. An air seal between the ear tip and the ear canal is required to change air pressure in the ear canal

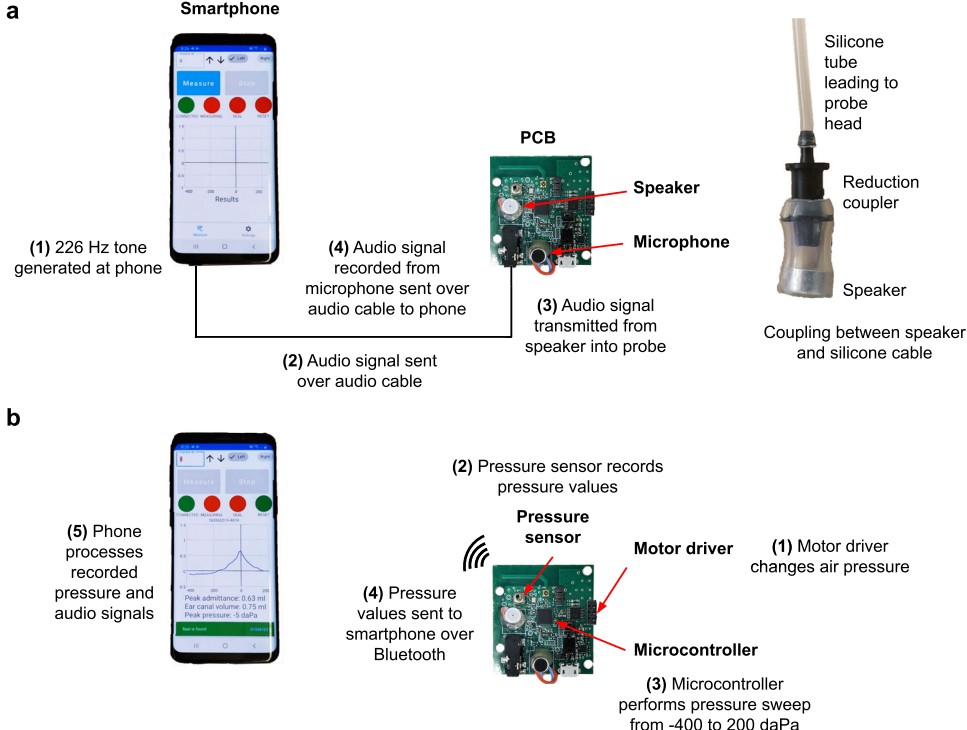

**Fig. 2 Working principle of our smartphone-based tympanometry device. a** Audio signal generation and reception. **b** Pressure sweeping, sensing and communications.

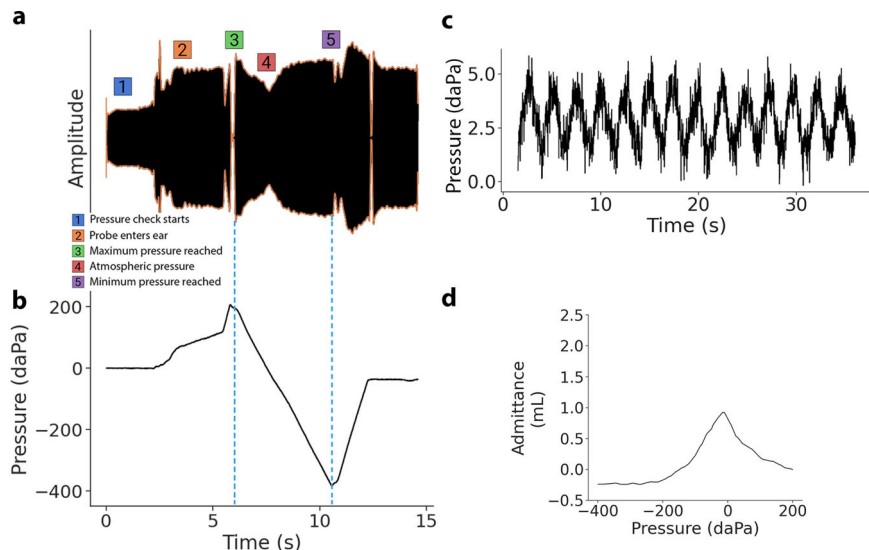

**Fig. 3 Measurement procedure to obtain tympanogram. a** Received acoustic signal bandpass filtered to 220--230 Hz. The acoustic signal increases in amplitude when it is inserted into the ear and changes in amplitude in response to air pressure changes. **b** Air pressure relative to atmospheric pressure as measured by the pressure sensor. The pressure signal spikes when the probe tip enters the ear. The pressure transducer performs a pressure sweep from 200 to −400 daPa, before returning to atmospheric pressure. **c** When the probe is outside the ear, only small pressure changes are recorded by the microcontroller. **d** Calibrated tympanogram in units of acoustic admittance magnitude.

space. Our seal detection algorithm has two steps. Firstly, we attempt to increase the pressure in the ear canal by moving the plunger of the syringe forward and backward. Specifically, we move the syringe plunger forward by 0.5 mm within an average time of 1.19 s. If the pressure sensor detects a sharp increase in pressure exceeding a threshold of 17 daPa/s, we record that a seal has been formed. When the probe is outside the ear, only a small change in pressure at an average rate of 2.03 daPa/s will be

registered (Fig. 3c). Even when there is no seal, a small change is still registered as the syringe introduces air pressure more quickly than air exiting the system. If no seal is detected, we move the syringe backwards by 0.5 mm. These steps are repeated until a seal is registered. We note that the pressure threshold for when a seal has been obtained has to be at least 17 daPa/s. We opted for a pressure threshold of 17 daPa/s as it was practical and yielded accurate measurements.

Relying on pressure changes alone to detect a seal can result in false seals from ear tip occlusions. For example, a false seal would occur if the ear tip were occluded with cerumen. In our second step, we track the amplitude of the reflected audio tone on the smartphone to detect occlusions (Supplementary Fig. 4). When there is an occlusion the amount of sound reflected back to the microphone is much higher than if the ear tip was clear. If the amplitude of the tone exceeds a predefined threshold of 65 dB SPL as recorded on the microphone, we inform the user that the ear tip is occluded. If there is no occlusion, the smartphone sends a Bluetooth notification to the microcontroller, which records that a seal has been formed. This threshold can be calibrated for different smartphones and probes by coupling the probe head to a 5 mL syringe with the plunger set to 0 mL as in Supplementary Fig. 5a, starting a measurement, and noting the acoustic amplitude value as received on the smartphone.

**Measurement procedure.** When a seal is obtained, the measurement begins by increasing the pressure in the ear canal to 200 daPa, performing a pressure sweep from 200 to −400 daPa, and then returning the ear canal back to the atmospheric pressure.

*Identifying when to start a measurement.* When the probe enters the ear canal and has an airtight seal, the pressure within the tubing system increases sharply. This is due to the pressure applied to the ear tip when it enters the ear canal, which causes air from the ear canal to suddenly enter the tubing system (Fig. 3b). If this initial pressure increase is below 200 daPa, the motor moves the syringe forward until the pressure in the ear reaches 200 daPa before starting the measurement. If the initial pressure increases beyond 200 daPa, the system starts the measurement and decreases the pressure to −400 daPa.

The initial pressure increase to 200 daPa, and later pressure increase from −400 daPa back to atmospheric pressure (0 daPa) is performed at a higher speed of $316 \pm 120$ daPa/s, compared to the speed of $123 \pm 18$ daPa/s during the pressure sweep. This was done to reduce measurement time.

*Performing the measurement.* During the measurement, the syringe is pulled backwards to sweep the pressure in the ear canal from 200 to −400 daPa. The motor speed is slowed to $123 \pm 18$ daPa/s during the measurement phase.

To ensure the plunger does not reach the end of the syringe, the session aborts if the syringe has moved forward by more than 16 mm during a measurement session. This scenario can occur if the ear tip loses its seal with the ear canal multiple times in a single measurement session.

*Completing the measurement.* After the pressure sweep is completed, the pressure in the ear canal is returned to atmospheric pressure. The user is then instructed to remove the ear tip from the ear canal. After 5 seconds, the syringe plunger is automatically moved back to its original position.

**Algorithm to compute tympanogram.** Our algorithm to compute the tympanogram ran in realtime on the smartphone during the clinical study and the tympanogram was displayed on the smartphone. The average runtime was $289 \pm 48$ ms across 10 measurements.

*Synchronizing pressure and audio streams.* In order to generate the tympanogram, pressure values recorded on the microcontroller need to be synchronized with the audio recording stored on the smartphone. This is challenging, as the clocks between the microcontroller and smartphone are not synchronized. The time shifts between the pressure and audio data are estimated using two steps. In the first step, the microcontroller sends a Bluetooth notification to the smartphone when the measurement is at the beginning and end of the pressure sweep (point 3 and 5 in Fig. 3a). The smartphone then records a timestamp when it has received these notifications. However, these timestamps do not account for the delay caused by transmitting and receiving the Bluetooth notification. As a result there is some unknown offset between the arrival of the notifications, and the beginning and end of the pressure sweep. In the second step, we estimate when the pressure sweep starts (point 3). To do this, we process the envelope of the audio recording starting from 500 ms before the Bluetooth notification to when the Bluetooth notification arrives. We observe that when there is a change in pressure direction, this causes the acoustic signal to dip for a brief period as shown in Fig. 3a. The maximum peak between this dip and the Bluetooth notification corresponds to the point of maximum pressure (point 3). This procedure is similarly repeated to find the point of minimum pressure (point 5). See Supplementary Algorithm 1 for a more detailed pseudocode block.

*Calculating tympanogram.* The goal of this step is to convert the synchronized pressure and audio data into a tympanogram. To do this, the audio data is divided into N segments, where N is the number of recorded pressure values between 200 and −400 daPa. For each audio segment, we perform a 24000-point FFT, and track the absolute value of the 226 Hz tone across all segments.

*Smoothing tympanogram.* The purpose of this step is to smooth the shape of the tympanogram and reduce the effect of noise. This is a two-part procedure. In the first part we discretize the admittance values into pressure bins of 5 daPa. Empty buckets are filled with the average admittance values in adjacent buckets. In the second part, we pass the tympanogram through a moving average filter with a window size of 5 samples.

**Calibration procedure.** A one-time calibration procedure needs to be performed prior to using the system. As the smartphone does not have the ability to measure the sound level at the microphone in absolute physical units of dB SPL, we perform sound level calibration using a sound level meter, which is able to perform this measurement. To do this, we first couple the probe head to a sound level meter (BAFX 3370, Digital Sound Level Meter, $18). We then adjust the volume gain of the speaker through the smartphone UI until the sound level output from the probe head reaches 85 dB SPL (Supplementary Fig. 6). We next calibrate the smartphone so the generated tympanograms can be normalized to absolute units of admittance. This calibration procedure is performed by measuring test cavities with volumes ranging from 0 to 5 mL in increments of 1 mL. For the calibration procedure, we use a 5 mL syringe with a diameter of 12.5 mm as our test cavity. Supplementary Fig. 5 shows the uncalibrated and calibrated flat tympanograms for a volume range of 0 to 5 mL. Calibration is performed by fitting the measured values at 200 daPa for all volumes to a cubic curve. The cubic fit produces four coefficients $p_1, p_2, p_3, p_4$ that are used to normalize an uncalibrated amplitude value $x$: $p_1 x^3 p_2 x^2 + p_3 x + p_4$. We note that the measured tympanograms exhibit a slope as pressure decreases. We compensate for this slope when computing the root-mean-square error of the tympanograms. To do this, we measure the slope $m$ of the tympanogram measured in the 0 mL configuration between −400 and 200 daPa, and the amplitude value $b$ at −400 daPa. We then created a line of best fit $y = mx + b$ where $x$ is vector of measured pressure values, and subtract this from all subsequent tympanogram measurements to compensate for the slope.

For the calibration procedure, the volume range of 0 to 5 mL is selected to match the recommended calibration cavity volumes of 0.5, 2, and 5 mL as specified in ANSI S3.39[20]. Additionally, we note that a meta-analysis[6] of several tympanometry studies show that the 90% range for ear canal volume ranges from 0.3 to 2.2 mL, and falls within the 0 to 5 mL range of calibration volumes. While our system is calibrated for the volume range of 0 to 5 mL, which would result in a plunger displacement of 5.3 to 7.1 mm, our system can accommodate a maximum plunger displacement of 16 mm of the entire reserve.

**Statistics and Reproducibility**. Algorithms to compute the tympanogram and its associated clinical metrics were calculated in real time on the Android platform. NumPy was used to perform correlation and Bland-Altman analysis and to calculate sensitivity, specificity and 95% CI values. Line charts were created using matplotlib and seaborn.

**Clinical study design**. All patients in the study were scheduled for an appointment with an audiologist and were undergoing tympanometry as part of their clinical visit. A licensed audiologist first performed tympanometry on subjects using our smartphone system and then with a commercial tympanometer (Supplementary Movie 1). The study was performed by two independent audiologists. The first audiologist tested 31 patient ears and used the GSI TympStar Pro as the reference tympanometer. The second audiologist tested 19 patient ears and used the GSI TympStar as the reference tympanometer. Otoscopy was performed prior to the measurement to check for the presence of ear tubes, eardrum abnormalities or cerumen obstruction. A behavioral audiogram was performed on each patient as part of their clinical visit, and any hearing loss, history of ear infection or relevant otologic conditions were recorded. The audiologist performing the test classified the tympanograms measured from the commercial tympanometer into Liden and Jerger classifications[21–24] using clinical criteria from our institution (Supplementary Table 4). All clinical tests were performed using the Samsung Galaxy S9 smartphone.

The study was approved by the Seattle Children's Hospital Institutional Review Board (STUDY00002820). All studies complied with relevant ethical regulations. Parental permission was obtained for participants under the age of 18 years. Children age 7 to 17 provided written assent. Assent was obtained after parental permission was granted. Children age 7 to 12 signed a simple assent form and children age 13 to 17 signed a consent form. Parents co-signed the consent form. Participants 18 years and older signed a consent form. The measured pressure speed was on average 125 ± 19 daPa/s in ears tested in the clinical study. The GSI TympStar and GSI TympStar Pro both had a pressure speed of 200 daPa/s. Recruitment lasted until a sufficient number of measurements were collected to demonstrate proof of concept. Randomization was not applicable, patients were not allocated into different groups. Investigators were not blinded.

**Reporting summary**. Further information on research design is available in the Nature Research Reporting Summary linked to this article.

## Results
**Clinical testing results**. We conducted a clinical study at Seattle Children's Hospital on 50 patient ears. The age range of the patients was 1 to 20 years with a mean age of 9 ± 5 years, and a female-to-male ratio of 0.52 (Supplementary Table 3). Of the 50 tested ears, 40 of them were classified as Type A by the audiologist, 5 as Type B, 3 as Type As, 1 as Type Ad, and 1 as Type C.

Of the 5 patient ears that were classified as producing Type B tympanograms, 4 had patent ear tubes and 1 had an eardrum perforation. One patient ear with a Type As tympanogram had clinically diagnosed otosclerosis. 27 of the 50 patient ears had a history of hearing loss, and 12 of the 50 ears had a history of ear infections. Figure 4 shows all the tympanograms classified as Type A, B, As, Ad and C by the audiologist, and the corresponding tympanogram obtained from our smartphone system.

Bland-Altman analysis[25] was performed on peak admittance values for the 45 ears without ear tubes (Fig. 5b). Since the presence of a patent ear tube would produce a flat tympanogram without a peak, peak admittance would not provide meaningful information about eardrum mobility. Bland-Altman analysis had a bias error (mean of the differences) of −0.02 ± 0.14 mL. One of 45 measurements were outside the 95% limits of agreement. The mean and standard deviation of absolute error of peak admittance was 0.11 ± 0.09 mL (Supplementary Table 7). The root-mean-square error between the tympanograms measured on the smartphone and commercial tympanometer was on average 0.17 ± 0.11 mL across all 45 ears without ear tubes (Supplementary Table 8). As measurements on the commercial tympanometer were not always recorded to the minimum pressure of −400 daPa, the root-mean-square error is computed for the range of pressure values that were recorded on both the commercial and smartphone device.

Bland-Altman analysis was also performed on ear canal volume measured on the smartphone and commercial tympanometer (Fig. 5d). We show the ear canal volume for the 45 ears with an intact eardrum and the 5 ears without an intact eardrum. Measured ear canal volume is dependent on the insertion depth of the probe tip into the ear canal, and may vary across measurements on the same device. Clinically, ear canal volume is primarily used in conjunction with tympanogram shape to screen for intact eardrums, or an occlusion of the probe tip or ear canal. The ear canal volumes measured by the smartphone for the 5 ears without an intact eardrum all exceeded 1.5 mL and appeared as flat tympanograms. Bland-Altman analysis for the full set of 50 ears showed a bias error of 0.15 ± 0.96 mL, and 2 of 50 measurements fell outside the limits of agreement. The same analysis on the 45 ears with an intact eardrum had a bias error of −0.09 ± 0.25 mL, and 1 of 45 measurements falling outside the limits of agreement. Mean and standard deviation of absolute error of ear canal volume was 0.41 ± 0.88 mL for the full set of 50 ears, and 0.20 ± 0.18 mL for the 45 ears (Supplementary Table 9). The ear canal volumes for all ears without an intact eardrum ranged from 1.61 to 3.97 mL and were larger than all the ear canal volumes of ears with an intact eardrum.

Finally, Bland-Altman analysis was performed for peak pressure in the 45 ears with an intact eardrum (Fig. 5f). In our clinical study, 1 ear had negative middle ear pressure of -225 daPa. Bland-Altman analysis showed a bias error of − 1 ± 17 daPa and a mean and standard deviation of absolute error of 13 ± 11 daPa (Supplementary Table 10). Four of 45 measurements fell outside the limits of agreement. Prior work[26] comparing the agreement in peak pressure between two commercial tympanometers reported a bias error of −16 ± 11 daPa.

**Tympanogram classification**. At the completion of the clinical study, we presented a total of 100 tympanograms measured on the smartphone and clinic device to a group of five pediatric audiologists and asked them to classify the tympanograms into Liden and Jerger classes (A, As, Ad, B, and C) based on the clinical criteria in Supplementary Table 4. The 100 tympanograms were anonymized and presented in a randomized order to each audiologist. Since the cutoffs for different tympanogram

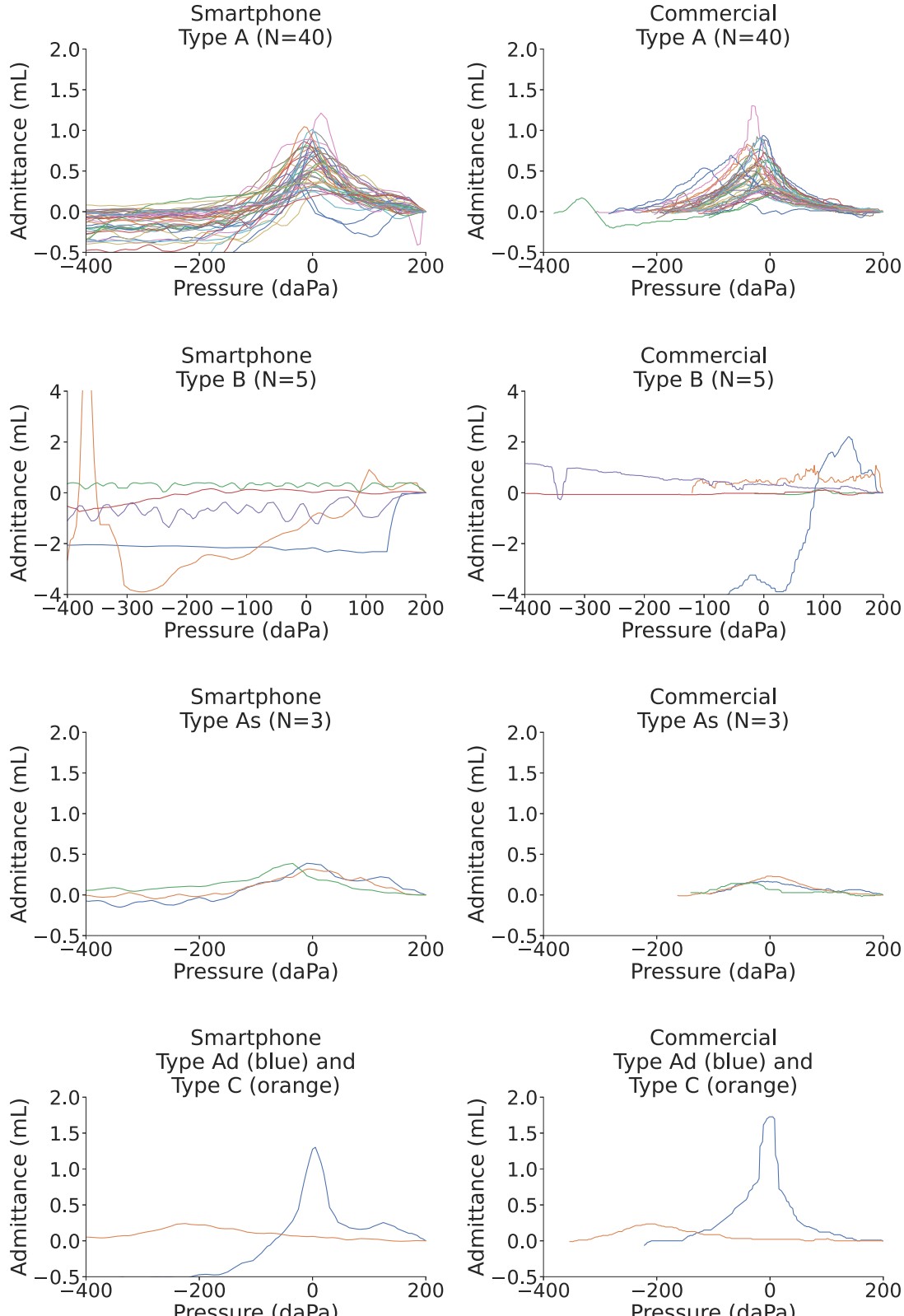

**Fig. 4 Comparison of individual tympanograms from smartphone-based and commercial tympanometer.** We show the plots for 40 type A, 5 type B, 3 type As, 1 type Ad, and 1 type C tympanograms measured by the commercial tympanometer, and tympanograms from the same ear as obtained by the smartphone-based tympanometer. Source data are provided in Supplementary Data 1.

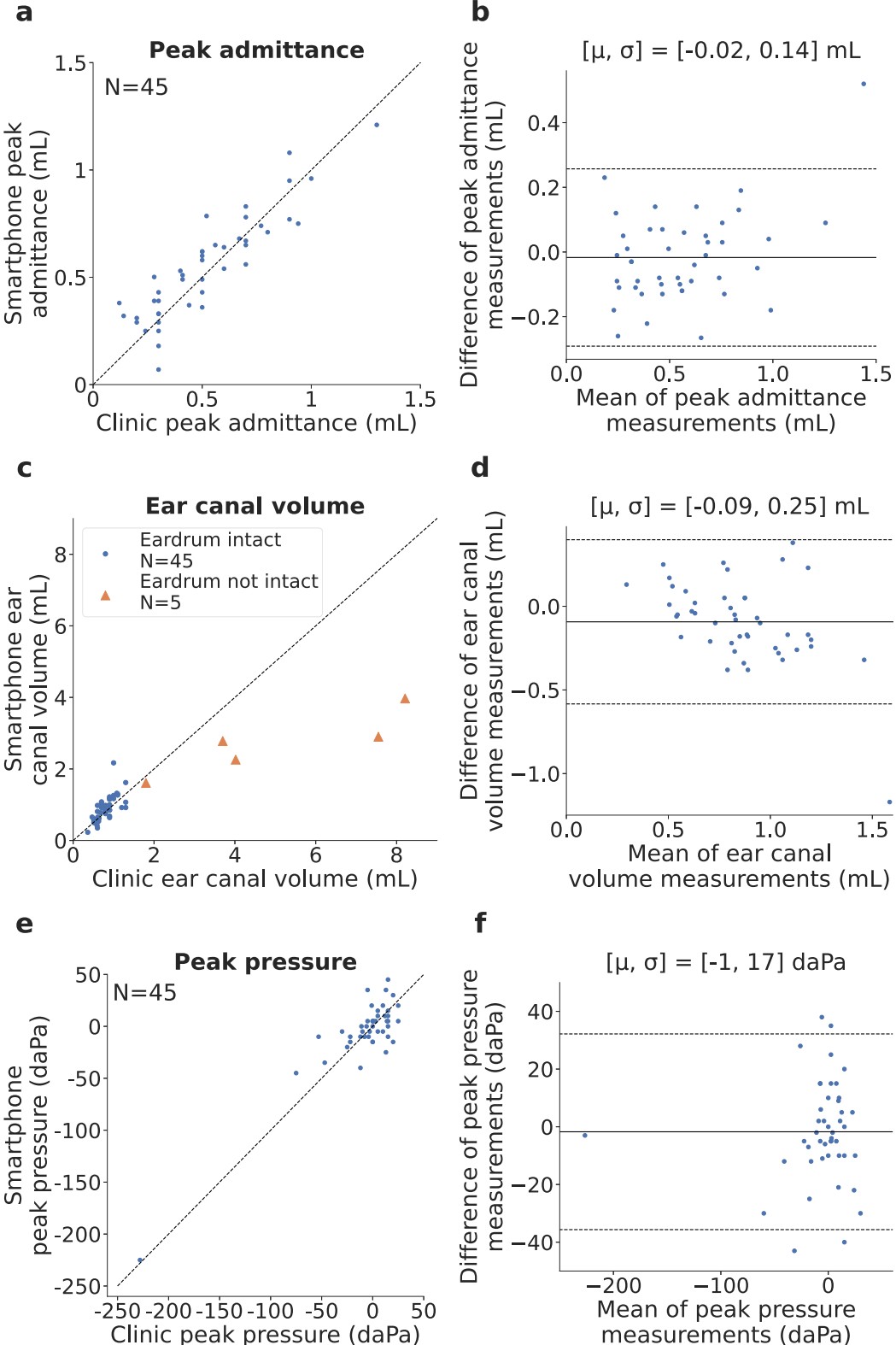

**Fig. 5 Clinical study performance. a**, **c**, **e** Correlation and **b**, **d**, **f** Bland-Altman plots compare peak admittance, ear canal volume and peak pressure obtained from the commercial tympanometer and smartphone-based tympanometer. In the Bland-Altman plot $\mu$ indicates bias error (mean of the differences) between measurements and $\sigma$ is the standard deviation (SD) of measurement error. The solid and dotted lines represent the bias error and 95% limits of agreement respectively. Source data are provided in Supplementary Data 1.

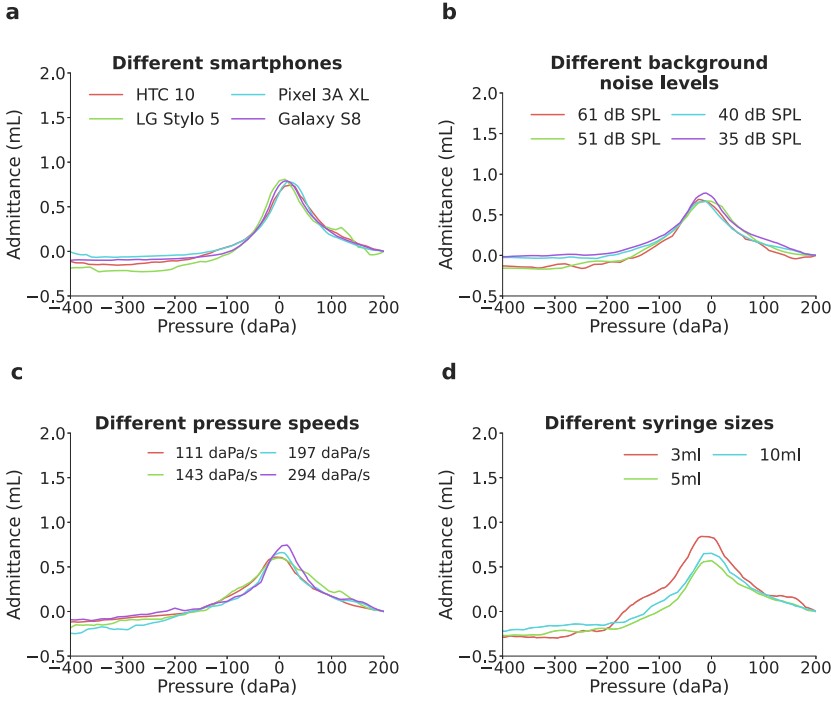

**Fig. 6 Benchmark testing across different measurement scenarios and design parameters.** The system was evaluated across **a** different smartphones including budget models, **b** different background noise levels **c** different pressure sweep speeds, **d** different syringe volumes. Source data are provided in Supplementary Data 1.

classes depend on the patient's age, we also provided the age information. The results of otoscopy, any prior ear surgery, and otologic history were also presented. The average percent agreement between both devices was on average $86 \pm 2\%$ across all five audiologists. Additionally, all type B tympanograms measured on the smartphone were correctly classified by the five participating audiologists.

Under the five-class taxonomy, there were ten tympanograms where disagreement was observed and the smartphone classification differed from the ground truth. Eight of the ten disagreements were for tympanograms with a ground truth of Type A or As, but were incorrectly classified as a Type A, As, or Ad. In seven of these cases, the smartphone tympanogram had a peak admittance that was 0.01 to 0.12 mL away from the threshold that would have caused it to be correctly categorized in the same class as the commercial tympanogram. A 0.12 mL peak admittance variation is within the test-retest variability for both the smartphone and commercial device (see Supplementary Note 1 and Supplementary Table 5).

**Benchmark testing**. We conducted benchmark testing across different variations in system design (Fig. 6, Supplementary Fig. 12, Supplementary Table 5, 6). Each benchmark test was performed in a single normal and healthy adult ear without history of middle ear disorders. The same ear was used for all tests. The default setup for these benchmark tests matched the setup used in the clinical study and used the Samsung Galaxy S9 and 5 mL syringe.

We first evaluated the effect of different smartphones on tympanogram output (Fig. 6a). The phones we tested were released during 2016 to 2019 and included smartphones that could be purchased second-hand for $50. We note that identical volume settings on each phone's software will result in slightly different sound levels at the speaker. This may be due to differences in the circuit driving the speaker. Across these phones,

the same volume setting resulted in sound levels ranging from 78 to 90 dB SPL. Our calibration procedure was performed for each phone in order to generate comparable tympanograms. We find that across all phones, the peak admittance and peak pressure was in the range of 0.74–0.81 mL and 10–20 daPa respectively.

Second, we tested the effect of interfering background noise on the tympanogram (Fig. 6b). We play an acoustic chirp from 220 to 230 Hz with a duration of 1 s in a continuous loop, using an external speaker (Beats Pill 2.0) positioned 50 cm away from the ear. The sound level of the acoustic chirp was measured at the microphone positioned at the end of the probe cable by coupling a sound meter (Amprobe SM-10) to the silicone tube that would attach to the microphone. The measured sound levels ranged from 35 to 61 dB SPL. The sound levels when measured next to the ear being measured ranged from 43 to 77 dB SPL. The sound levels when measured 1 cm away from the speaker ranged from 75 to 110 dB SPL. The tympanogram is slightly less smooth at the highest sound level tested, but otherwise has no appreciable effect on the overall shape of the tympanogram. Peak admittance ranged from 0.67 to 0.77 mL and peak pressure ranged from −25 to −10 daPa across all sound levels.

Third, we tested how different pressure speeds would affect the tympanogram (Fig. 6c). Pressure speed was varied by changing the rate at which the stepper motor pulled the syringe plunger. We tested four different speeds ranging from 111 to 294 daPa/s. These speeds correspond to a pressure sweep duration of 2.04 to 5.45 s. Across different speeds the peak admittance ranged from 0.60 to 0.74 mL and peak pressure ranged from −15 to 5 daPa. Additionally, we tested the effects of different pressure speeds on the commercial tympanometers (GSI TympStar and GSI TympStar Pro) used in the clinical study, and measured a healthy adult ear three times. Both machines had preset measurements speeds of 12.5, 50, and 200 daPa/s. On the GSI TympStar, peak admittance ranged from 0.60 to 0.70 mL while peak pressure ranged from − 10 to 5 daPa. On the GSI TympStar Pro, peak

admittance ranged from 0.60 to 0.68 mL while peak pressure ranged from −41 to 0 daPa.

Fourth, we show the effect of different syringes volumes on the tympanogram (Fig. 6d). We tested three different syringe volumes ranging from 3 to 10 mL, with syringe diameter increasing from 10 to 16 mm. While the stepper motor moved at the same speed for these experiments, the use of larger syringes resulted in higher pressure speeds due to the increased diameter of the plunger. Pressure speeds ranged from 79 to 150 daPa/s across the three syringe volumes. Across all syringes the peak admittance ranged from 0.57 to 0.65 mL and peak pressure ranged from −5 to 0 daPa.

Fifth, we show the admittance values when the ear probe is connected to hard-backed cavities of varying volume. Our cavity is a 5 mL syringe, with the plunger retracted at different volume levels (Supplementary Fig. 5a). We can see that a flat tympanogram is produced at each volume level from 0 to 5 mL. The average root-mean-square error of the calibrated tympanograms was $0.09 \pm 0.02$ mL across all volumes (Supplementary Fig. 5e).

Finally, to evaluate if untrained participants could perform the calibration procedure, we provided two independent participants with written instructions and diagrams (Supplementary Fig. 13) for performing the process and compared their calibration results with that of a trained researcher. Specifically, the device was first calibrated to the measurements obtained by the trained researcher, and the obtained cubic coefficients were used to normalize measurements for the two independent participants. No additional calibration was performed for the two participants. Curves from all participants were compared against the ground truth syringe volume. The average root-mean-square error across the pressure range of −400 to 200 daPa, for all volumes from 0 to 5 mL was 0.08 mL for the trained researcher and 0.08, and 0.07 mL for the independent participants. The independent participants were able to read the instructions and perform the entire calibration process in 4:13 and 4:35 minutes.

**Discussion**. Existing techniques to measure middle ear function leverage different sensing techniques to probe the mobility of the eardrum and middle ear system.[27] leverages the speakers and microphones on smartphones to assess eardrum mobility using acoustic reflectometry. Unlike prior work, the smartphone-based tympanometer presented here provides similar information to commercially available tympanometry, and can be used to aid clinical diagnosis of a variety of pathologies affecting the middle ear. KUDUwave TMP[28] is a recent commercial device that integrates two pneumatic pumps into circumaural ear cups and ear inserts to perform bilateral tympanometry and audiometry and costs $6950. Our system mimics traditional unilateral tympanometry and is built on a smartphone device in both a handheld and desktop form factor. Visual otoscopy is often performed prior to tympanometry to screen for a bulging or cloudy tympanic membrane or an ear canal occluded with cerumen[29]. Visual otoscopes are also readily available as smartphone attachments[30–32], but are known to have poor sensitivity and specificity at detecting middle ear disorders[2,33,34].

Optical coherence tomography (OCT) uses the reflections of infrared light to try to gain information about the middle ear[35–37]. While OCT devices are available on the market, they are not widely adopted. Unlike these tests, our smartphone-based tympanometry is able to provide information about middle ear function similar to commercial tympanometry. The portable nature of our system makes it a potentially viable option for use in developing countries, primary care clinics and mobile health clinics.

Looking forward, our system was tested in a pediatric population with a minimum age of one year with a tone frequency of 226 Hz. Due to known developmental differences in mass and stiffness characteristics of the outer and middle ears of infants age 0 to 9 months[38,39], tympanometry at 1000 Hz is recommended[40,41]. Further research is needed to evaluate the utility of our system in young infants. Furthermore, while our current design has been optimized and evaluated for tympanometry, a similar design could potentially be used to perform related tests including acoustic reflexes, which is often performed by commercial tympanometers in ipsilateral or contralateral mode. Related audiological tests that rely on the use of microphones and speakers such as audiometry, and otoacoustic emissions could potentially be integrated into a similar hardware design. Further work would be required to develop these designs and evaluate their clinical performance.

In summary, we present an open-source smartphone-based tympanometry system to measure middle ear function. Although commercial tympanometry is the standard of care to assess middle ear function, the test equipment is expensive and not readily available. Given the prevalence of inexpensive budget smartphones, particularly in developing countries, our frugal system has the potential to be a screening tool for middle ear disorders in resource-constrained environments. Further clinical studies are required to assess the technology's efficacy and acceptability for use in these environments.

### Data availability

All data supporting the findings from this study are available within the article and its supplementary information. Source data are provided as Supplementary Data 1 with this paper.

### Code availability

Code is available at https://github.com/uw-x/tymp[42].

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

## Acknowledgements

We thank our participants and their families at Seattle Children's Hospital for their willingness to participate in our study. We would like to thank the Department of Audiology at Seattle Children's Hospital, in particular R. Castillo for enabling and facilitating patient recruitment. We thank J. Dunnell for her critical and important feedback on the manuscript and methods. We are grateful for funding from the Bloedel Center Mini-Grant, Seattle Children's Research Institute, Research Integration Hub, Pilot Awards Support Fund Program, National Science Foundation, and Moore foundation fellowship.

## Author contributions

All authors designed the experiments, interpreted the results, and wrote the manuscript. J.C., A.N. conducted the experiments and performed the analysis with technical supervision by S.G., M.B., J.K., L.M., S.N., and R.B. M.B., J.K., L.M. collected data for the clinical study. J.C., A.N. and S.G. designed the algorithms. J.C., A.N., and S.G. conceptualized the study.

## Competing interests

SG is a co-founder of Jeeva Wireless, Inc., and Sound Life Sciences, Inc.; SG, JC and RB are co-founders of Wavely Diagnostics, Inc. RB is a co-founder of EigenHealth Inc. RB is a consultant and stock holder of Spiway, LLC. The remaining authors declare no competing interests.
