## [Peer Review File · Communications Medicine]

Reviewers' comments:

Reviewer #1 (Remarks to the Author):

This is an interesting paper & study, well designed and validated (related to the 2256 Hz frequency). Although the objective of the authors is to present the 226 Hz data, the paper would benefit significantly if the authors could comment on the technical issues related to higher frequencies used in pediatric Tympanometry, such as 1000 Hz.

Reviewer #2 (Remarks to the Author):

This manuscript describes the development and validation of a portable tympanometer designed for use with a smartphone.

The authors show that the device is comparable to a commercial tympanometer and provide information on how to build such a device as they want this to be open-source.

Considering the size of commercial tympanometers, this study is innovative in demonstrating the feasibility and accuracy of smartphone based tympanometers.

I do think it is worth publishing as it will be provocative to many people who are interested in bringing tympanometry to remote areas, or any area where it is difficult to bring larger equipment.

My only question is the use of audiologists to characterize the results by the Jerger types. There are normative values that should be used to classify the Jerger type. Why not use those norms to classify the tympanograms from the new and commercial equipment and then see how closely they match?

Reviewer #3 (Remarks to the Author):

The manuscript presents a low cost tympanometer that is benchmarked against the clinical gold standard. The device shows agreement with the gold standard in a clinical trial with many patients and a variety of conditions.

The work is significant as it potentially enables distribution of a low cost tympanometer to resource scarce populations.

The manuscript presents the engineering content at a relatively shallow level, there are several materials missing to be able to reproduce the device and more in depth discussions about component choice and the electrical circuit should be included to support the open source nature of the system.

Specific comments:

- The manuscript could benefit from better graphics and or description of the working principle of the device. It is for example not clear where the audio signal is generated and why cabled connections to a smartphone are needed in the first place.
- More detail is needed for this statement: "To ensure safety, our system is designed to terminate the measurement if the syringe of diameter 12.5 mm has moved forward by more than 16 mm."

How is this related to safety?

- There is no detail on the phones used in the test, the authors also report that the phones were calibrated to produce similar sound output levels. This is potentially a problem for deployment. More detail on the procedure is needed. Why can't the audio signal be created with the microcontroller? This would reduce variance between hardware.
- There is little information about how components on the PCB operate, I would strongly suggest to either put a simplified diagram in the main figures or add the full schematic and board layout to the SI and discuss function.
- The authors claim the design is open source but do not provide the code for the app and the code for the SOC the CAD files for the housing are also not provided. The schematic, board design and BOM is also missing.

We thank the editors and the reviewers for all of their insightful comments and constructive suggestions. Please find below a summary of the main changes:

1. Open-sourced all hardware designs (schematic and PCB files), bill of materials, code used for the microcontroller and smartphone.
2. Added supplementary figure illustrating principle of operation of our system.
3. Added benchmark showing the output of our system at 1000 Hz.
4. Clarified which smartphone was used for testing, and how the sound output levels are calibrated for use across different smartphones.

Reviewer #1 (Remarks to the Author):

This is an interesting paper & study, well designed and validated (related to the 2256 Hz frequency). Although the objective of the authors is to present the 226 Hz data, the paper would benefit significantly if the authors could comment on the technical issues related to higher frequencies used in pediatric Tympanometry, such as 1000 Hz.

Thank you for noting this, our system can be configured to emit different frequency tones from the smartphone including 1000 Hz. The output sound level of the 1000 Hz tone can be set to 85 dB SPL, which is the same sound level used for the 226 Hz in our main clinical study.

We have added a benchmark result in Supplementary Fig. 8 showing a 1000 Hz tympanogram as measured in a healthy adult ear without middle ear history. However, as we note in the discussion there are known developmental differences in mass and stiffness characteristics of the outer and middle ear of infants aged 0 to 9 months. Further testing is required to evaluate the utility of our system in young infants.

We have described this benchmark experiment in the Supplementary Information (Line 55-59): *Our system can be configured to emit different frequency tones in software including at 1000 Hz. The 1000 Hz tone was calibrated to transmit at 85 dB SPL, which is the same sound level output as used for the 226 Hz tone in our clinical study. We measured the tympanogram at 1000 Hz in a single healthy adult ear without prior middle ear history in Supplementary Fig. 8. We note that further testing is required to evaluate the utility of our system in young infants.*

Reviewer #2 (Remarks to the Author):

This manuscript describes the development and validation of a portable tympanometer designed for use with a smartphone.

The authors show that the device is comparable to a commercial tympanometer and provide information on how to build such a device as they want this to be open-source.

Considering the size of commercial tympanometers, this study is innovative in demonstrating the feasibility and accuracy of smartphone based tympanometers.

I do think it is worth publishing as it will be provocative to many people who are interested in bringing tympanometry to remote areas, or any area where it is difficult to bring larger equipment.

My only question is the use of audiologists to characterize the results by the Jerger types. There are normative values that should be used to classify the Jerger type. Why not use those norms to classify the tympanograms from the new and commercial equipment and then see how closely they match?

Thank you for noting this. To reflect the real-world clinical procedure followed by the audiologists as part of their practice, we designed the study so that the tympanogram classification into Jerger type was performed by the audiologist using the normative clinical criteria prescribed by our institution (Supplementary Table 1):

	Adults (11 years and up)	Children (6 months - 10 years)
Static compliance	0.3 to 1.4 mL (mean 0.8)	0.2 to 0.9 mL (mean 0.5)
Ear canal volume	0.6 to 1.5~mL	0.3 to 0.9~mL
Middle ear pressure	-110 to 150 daPa	-150 to 150 daPa

Tympanometry types: (Liden and Jerger classifications)

1. **Type A:** normal "A or mountain" shape
2. **Type B:** flat tracing non-measurable static compliance
3. **Type C:** negative middle ear air pressure
 - a. Children 6 months to 10 years: peak pressure less than -150 daPa
 - b. 11 years of age and up: peak pressure than -110 daPa
4. **Type As:** height is decreased or shallow (i.e. otosclerosis or fluid), compliance 0.2 mL or 0.1 mL
5. **Type Ad:** height is significantly increased of deep (i.e. ossicular disarticulation)
 - a. Children 6 months to 10 years: compliance 1.0 mL or greater
 - b. 11 years of age and up: compliance 1.5 mL of greater

Supplementary Table 1: Clinical criteria for tympanogram classification. Criteria and normative tympanometry values used by audiologists at our institution for classifying tympanograms into Liden and Jerger classifications.

Furthermore we note that we also performed a quantitative comparison of the tympanogram's peak admittance, ear canal volume, and peak pressure values in the 'Clinical testing results' section and in Fig. 5:

Figure 5: Clinical study performance. a–f, Correlation (left-column) and Bland-Altman (right column) plots compare peak admittance, ear canal volume and peak pressure obtained from the commercial tympanometer and smartphone-based tympanometer. In the Bland-Altman plot μ indicates bias error (mean of the differences) between measurements and σ is the standard deviation (SD) of measurement error. The solid and dotted lines represent the bias error and 95% limits of agreement respectively.

Reviewer #3 (Remarks to the Author):

The manuscript presents a low cost tympanometer that is benchmarked against the clinical gold standard. The device shows agreement with the gold standard in a clinical trial with many patients and a variety of conditions.

The work is significant as it potentially enables distribution of a low cost tympanometer to resource scarce populations.

The manuscript presents the engineering content at a relatively shallow level, there are several materials missing to be able to reproduce the device and more in depth discussions about component choice and the electrical circuit should be included to support the open source nature of the system.

Thank you for the positive comments. Please see the notes below about engineering material.

Specific comments:

- The manuscript could benefit from better graphics and or description of the working principle of the device. It is for example not clear where the audio signal is generated and why cabled connections to a smartphone are needed in the first place.

We have added Fig. 2 to further illustrate the working principle of the device.

The figure shows that the audio signal is generated by the smartphone. The audio cable is used to connect the smartphone to the speaker on the PCB. We used a custom speaker as it was easier to create an airtight seal coupling it to the silicone cable.

Fig. 2. Working principle of our smartphone-based tympanometry device. a, Audio signal generation and reception. b, Pressure sweeping, sensing and communications.

- More detail is needed for this statement: “To ensure safety, our system is designed to terminate the measurement if the syringe of diameter 12.5 mm has moved forward by more than 16 mm.” How is this related to safety?

We have clarified and modified the line as follows:

This termination condition was added to avoid large changes in air pressure in the ear. It is a fail-safe in the event of malfunction with the pressure sensor. We note that during our study there were no failures with the pressure sensor. (Line 64-66)

- There is no detail on the phones used in the test, the authors also report that the phones were calibrated to produce similar sound output levels. This is potentially a problem for deployment. More detail on the procedure is needed. Why can't the audio signal be created with the microcontroller? This would reduce variance between hardware.

We have added clarification to the Study Design that the phone used for all the clinical testing was a Samsung Galaxy S9:

All clinical tests were performed using the Samsung Galaxy S9 smartphone. (Line 110)

The default setup for these benchmark tests matched the setup used in the clinical study and used the Samsung Galaxy S9 and 5 mL syringe. (Line 113-114)

We have added further detail in the Calibration Procedure section and Supplementary Fig. 5 on how the sound level output is calibrated for different smartphones:

A one-time calibration procedure needs to be performed prior to using the system. As the smartphone does not have the ability to measure the sound level at the microphone in absolute physical units of dB SPL, we perform sound level calibration using a sound level meter which is able to perform this measurement. To do this, we first couple the probe head to a sound level meter (BAFX 3370, Digital Sound Level Meter, \$18). We then adjust the volume gain of the speaker through the smartphone UI until the sound level output from the probe head reaches 85 dB SPL (Supplementary Fig. 5). (Line 381-386)

Supplementary Figure 5: Sound level calibration. A sound meter (BAFX 3370, Digital Sound Level Meter, \$18) is used to calibrate the sound level produced by our system.

We opted to use the smartphone to generate audio signals, as it already had a built-in capability to perform that function and was convenient to program. In our benchmark evaluation, we show that after a one-time calibration, different smartphones produce comparable tympanograms.

Further, we envision that in a deployment scenario, measurements would be performed using a dedicated smartphone which would have already been calibrated to produce sound at a specific output level, and display tympanograms in absolute units of admittance.

Fig. 6: Benchmark testing across different measurement scenarios and design parameters. The system was evaluated across different smartphones including budget models.

We first evaluated the effect of different smartphones on tympanogram output (Fig. 6a). The phones we tested were released during 2016 to 2019 and included smartphones that could be purchased second-hand for \$50. We note that identical volume settings on each phone's software will result in slightly different sound levels at the speaker. This may be due to differences in the circuit driving the speaker. Across these phones, the same volume setting resulted in sound levels ranging from 78 to 90 dB SPL. Our calibration procedure was performed for each phone in order to generate comparable tympanograms. We find that across all phones, the peak admittance and peak pressure was in the range of 0.74 to 0.81 mL and 10 to 20 daPa respectively.

- There is little information about how components on the PCB operate, I would strongly suggest to either put a simplified diagram in the main figures or add the full schematic and board layout to the SI and discuss function.

We have added the schematics to Supplementary Fig. 9 to 13 and described the role of each schematic diagram in the caption:

Supplementary Fig. 9: Schematic of microcontroller connections.

Supplementary Fig. 10: Schematic of motor driver connections.

Supplementary Fig. 11: Schematic of power supply connections.

Supplementary Fig. 12: Schematic of pressure sensor connections.

Supplementary Fig. 13: Schematic of speaker and microphone connections.

We have also uploaded the full design of our system including the hardware design (schematics, board layout, raw PCB design files) to our public Github repository under open source license (Apache License 2.0): <https://github.com/uw-x/tymp>

- The authors claim the design is open source but do not provide the code for the app and the code for the SOC the CAD files for the housing are also not provided. The schematic, board design and BOM is also missing.

We have uploaded the full design for our system to a public Github repository under an open source license (Apache License 2.0):

<https://github.com/uw-x/tymp>

The repository contains the hardware design (schematic and PCB design files), bill of materials, and code for the microcontroller and smartphone app.

We have added a link to this repository in the Data Availability section of the paper.

We have specified the manufacturer, part number, and dimensions of the plastic housing that encloses our system on line 292-294:

*All the components in the desktop version of our system are enclosed in a plastic enclosure
(Hammond Manufacturing 1591XXFSFL, 221.01 x 150.01 x 63.50 mm)*

REVIEWERS' COMMENTS:

Reviewer #1 (Remarks to the Author):

The revised version addresses the topic of 1000 Hz, therefore the paper is suitable for publication.

Reviewer #2 (Remarks to the Author):

Thank you for resubmitting your manuscript and addressing the concerns brought up previously. This article provides instruction for the construction of and support for use of a smart-phone tympanometer. This is highly innovative - I would not have thought it possible to create such a product. I believe these instructions would be helpful in bringing this important medical procedure to areas with limited resources. I recommend accepting this manuscript.

Reviewer #3 (Remarks to the Author):

The Authors have addressed the reviewers comments in detail and the manuscript can be published as is.